# Radiomics Analysis of Whole-Kidney Non-Contrast CT for Early Identification of Chronic Kidney Disease Stages 1–3

**DOI:** 10.3390/bioengineering12050454

**Published:** 2025-04-25

**Authors:** Guirong Zhang, Pan Zhang, Yuwei Xia, Feng Shi, Yuelang Zhang, Dun Ding

**Affiliations:** 1Department of Radiology, The Second Affiliated Hospital of Xi’an Jiaotong University, Xi’an 710004, China; guirjyhy@163.com (G.Z.); yangbujiwuyuan@126.com (P.Z.); zhangyuelang@mail.xjtu.edu.cn (Y.Z.); 2Department of Research and Development, United Imaging Intelligence, Shanghai 200232, China; yuwei.xia@uii-ai.com (Y.X.); feng.shi@uii-ai.com (F.S.)

**Keywords:** chronic kidney disease, radiomics, non-contrast computed tomography, estimated glomerular filtration rate, machine learning

## Abstract

Background: The early stages of chronic kidney disease (CKD) are often undetectable on traditional non-contrast computed tomography (NCCT) images through visual assessment by radiologists. This study aims to evaluate the potential of radiomics-based quantitative features extracted from NCCT, combined with machine learning techniques, in differentiating CKD stages 1–3 from healthy controls. Methods: This retrospective study involved 1099 CKD patients (stages 1–3) and 1099 healthy participants who underwent NCCT. Bilateral kidney volumes of interest were automatically segmented using a deep learning-based segmentation approach (VB-net) on CT images. Radiomics models were constructed using the mean values of features extracted from both kidneys. Key features were selected through Relief, MRMR, and LASSO regression algorithms. A machine learning classifier was trained to differentiate CKD from healthy kidneys and compared with the radiologist assessments. Model performance was evaluated using the area under the curve (AUC) of receiver operating characteristic analysis. Results: In the training set, the AUCs for the Gaussian process (GP) classifier model and radiologist assessments were 0.849 and 0.570, respectively. In the testing set, the AUC values were 0.790 for the GP model and 0.575 for radiologist assessments. Conclusions: The NCCT-based radiomics model demonstrates significant clinical utility by enabling non-invasive, early diagnosis of CKD stages 1–3, outperforming radiologist assessments.

## 1. Introduction

Chronic kidney disease (CKD) has become a significant public health issue worldwide, posing a serious threat to human health and contributing to a substantial increase in the burden of cardiovascular disease, other morbidity and mortality [1]. The estimated global prevalence of CKD in 2017 was 9.1% [2], and it is estimated to become the fifth leading cause of death globally by 2040 [3]. In a Chinese adult cross-sectional study conducted from 2018 to 2019, the estimated prevalence of CKD was 8.2%, affecting approximately 82 million adults. Among these individuals, 73.3% were classified at stages 1 and 2, 25.0% at stage 3, and 1.8% at stage 4 and 5. However, the awareness of CKD was found to be only 10.0% [4]. The estimated glomerular filtration rate (eGFR), calculated using the Chronic Kidney Disease Epidemiology Collaboration (CKD-EPI) equation, has become a critical basis for CKD staging [5]. Patients with CKD are classified into five progressive stages based on eGFR, and mortality increases with disease stage progression [6].

Unlike in developed countries, the proportion of patients in the early to intermediate stages of CKD in China is as high as 98.3%. Since early-stage CKD is often asymptomatic and frequently goes unrecognized in clinical practice, studies have shown that early diagnosis and intervention can effectively slow the progression of kidney damage [7]. Therefore, there is an urgent need to screen and identify early-stage CKD, as this is a critical step in halting or even reversing disease progression.

Non-contrast computed tomography (NCCT) is a non-invasive imaging technique with excellent spatial resolution and no risk of contrast-induced nephropathy, making it widely used for kidney disease screening. However, in the early stages of CKD, it is challenging to distinguish between healthy and diseased kidneys on CT images with the naked eye.

Radiomics leverages advanced algorithms to transform standard medical images into high-dimensional data arrays, enabling the identification of subtle structural changes in the kidneys that are beyond human perception. This approach represents a significant advancement over traditional imaging methods and aids in disease characterization. Compared to conventional CT examinations, CT radiomics can extract more detailed information about subtle lesions. It has been applied in various contexts, such as differentiating kidney stones based on NCCT [8], identifying renal tumors [9], and predicting radiation-induced kidney damage using contrast-enhanced CT [10].

Renal fibrosis characterizes virtually all progressive renal diseases [11]. This pathological process leads to microstructural changes in the kidneys, which may alter texture and other features compared to healthy kidneys. This study aims to investigate the diagnostic potential of radiomic features derived from NCCT images in distinguishing between CKD stages 1–3 and healthy kidneys.

## 2. Materials and Methods

### 2.1. Ethics Statement

This retrospective study was conducted with approval by the Institutional Review Board of The Second Affiliated Hospital of Xi’an Jiaotong University (Approval No. 2021168), and the requirement for informed consent was waived.

### 2.2. Selection of Study Participants

The inclusion criteria were as follows: patients admitted to the Department of Nephrology and diagnosed with CKD, aged ≥ 18 years, who underwent abdominal non-contrast CT scans and laboratory tests at the same time. The diagnosis of CKD was based on the KDIGO guidelines [12], and patients with CKD stages 1 to 3, as determined by eGFR, were included. Patients and age- and sex-matched healthy controls from January 2020 to September 2024 were retrospectively reviewed. Exclusion criteria included incomplete clinical or laboratory data, poor or incomplete image quality, large renal cysts (diameter > 3 cm), renal agenesis, asymmetric bilateral kidney atrophy, renal calculi, hydronephrosis, benign or malignant renal tumors, prior kidney biopsy, other urinary tract diseases, or a history of renal radiation therapy.

### 2.3. CT Imaging

All abdominal CT images were acquired using one of the following CT scanners: Somatom Definition Flash, Somatom go.Top, and Somatom Force (Siemens, Munich, Germany); Revolution CT and LightSpeed 64 (GE Healthcare, Chicago, IL, USA); uCT 780 (United Imaging, Shanghai, China). The CT scanning parameters were as follows: tube voltage, 120 kV; automatic tube current; matrix size, 512 × 512; and reconstructed slice thickness, 1 mm. The images were retrieved from the Picture Archiving and Communication System and uploaded to the research platform.

### 2.4. Kidney Segmentation

For the segmentation of both kidneys, the VB-net [13,14] kidney automatic segmentation algorithm was utilized. This algorithm was developed by United Imaging Intelligence’s one-stop research platform (uAI Research Portal, V20240730, https://urp.united-imaging.com/; accessed on 31 December 2024) [15]. Based on the differences in CT values between the renal sinus and renal parenchyma, the adipose tissue of the renal sinus was removed, retaining only the renal parenchyma. All segmentation masks were reviewed and verified by an experienced radiologist, with manual adjustments performed when necessary.

### 2.5. Radiomics Analysis

All CT images were resampled to a uniform voxel spacing of 1 × 1 × 1 mm^3^ using the B-spline interpolation algorithm, standardizing variable pixel sizes and slice thicknesses. This preprocessing step ensures that the machine learning model receives consistent input data, enhancing its ability to learn and generalize to new images. Following preprocessing, a total of 2264 radiomics features were automatically extracted from the volumes of interest (VOIs) of both kidneys for each patient.

These radiomics features included first-order statistics and shape and texture features. The texture features comprised Gray Level Size Zone Matrix, Gray Level Co-occurrence Matrix, Gray Level Run Length Matrix, Neighboring Gray Tone Difference Matrix, and Gray Level Dependence Matrix features. Additionally, 24 filters (Box Mean, Additive Gaussian Noise, Binomial Blur, Curvature Flow, Box-sigma, Normalize, Laplacian Sharpening, Discrete Gaussian, Mean, Speckle Noise, Recursive Gaussian, Shot Noise, LoG (sigma: 0.5, 1, 2, 4), and Wavelet (HHH, HLL, HLH, HHL, LLL, LLH, LHL, LHH)) were also applied to obtain the derived images. First-order statistics and texture features were then extracted based on the derived image.

### 2.6. Feature Selection and Prediction Model Establishment

The mean value of each feature from the right and left kidneys was calculated and used to construct radiomics models. For feature selection, each feature was normalized to standardized z-scores to minimize distortion in the differences among the radiomic features.

The data were randomly allocated to the training and testing sets at an 8:2 ratio. A three-step procedure was implemented to identify robust radiomic features within the training set. First, the Relief method was used for preliminary feature selection, retaining the top 50 features. Next, the Max-Relevance and Min-Redundancy (MRMR) algorithm was used to identify the feature sets with the strongest correlation and the least redundancy, retaining the top 30 features. Finally, the Least Absolute Shrinkage and Selection Operator (LASSO) regression analysis was employed to identify significant radiomic features with non-zero coefficients capable of distinguishing CKD patients from healthy controls.

The parameter α of LASSO was set to 0.05 to prevent overfitting. The final selected features were used to train a Gaussian process (GP) machine learning model to differentiate CKD patients from healthy controls.

### 2.7. Radiologist Diagnosis

Radiologist diagnostic classification was performed by two attending physicians, each with six years of experience. Any discrepancies were resolved by senior radiologists. All radiologists were blinded to clinical diagnosis, laboratory tests, and kidney biopsy pathological results. The diagnostic criteria for CKD were defined as structural abnormalities detected through imaging, typically manifested as renal atrophy or abnormal renal contours. The diagnosis was primarily based on clinical experience.

### 2.8. Statistical Analysis

The Shapiro–Wilk test was employed to assess the normality of the datasets. Continuous variables with a normal distribution are presented as mean ± standard deviation, whereas those with a non-normal distribution are reported as median (interquartile range). Categorical data are described as frequency counts (in percentages). For statistical comparisons, Student’s *t*-test was used for normally distributed continuous variables, the Mann–Whitney *U*-test was applied for non-normally distributed continuous variables, and the chi-square test was used for categorical variables to determine statistically significant differences. All statistical analyses were performed using Python 3.10.6. Two-sided tests were conducted, and *p*-values < 0.05 were considered statistically significant. The diagnostic performance was evaluated using the area under the receiver operating characteristic (ROC) curve. Additional metrics, including sensitivity, specificity, accuracy, and F1 score, were also calculated to assess model performance. The comparative performance between radiologists and radiomics model was determined using the DeLong test.

## 3. Results

### 3.1. Basic Clinical Information of Participants

The inclusion flowchart of patients and dataset distribution is shown in Figure 1. This study included a total of 1099 patients and 1099 healthy controls. Among the 1099 CKD patients, 534 were classified as stage 1, 237 as stage 2, and 328 as stage 3. These CKD patients exhibited a variety of etiologies: primary glomerular diseases primarily included chronic nephritis syndrome, nephrotic syndrome, IgA nephropathy, and membranous nephropathy, while secondary glomerular diseases included diabetes, hypertension, systemic lupus erythematosus, and other factors.

In the CKD group, 53.5% of participants were male, compared to 50.1% in the healthy control group. The median age of both the CKD patients and healthy controls was 50 years (interquartile range: 38, 60). No significant differences were observed in age or gender distribution within or between the training and testing sets (Table 1).

### 3.2. CT Radiomics Feature Selection, Model Construction

The workflow of radiomics analysis is illustrated in Figure 2. A total of 2264 radiomics features were extracted from the segmented right and left kidney. The mean value of each radiomic feature, calculated separately for the right and left kidneys, was used in the analysis. From these, 24 critical radiomics features were selected based on the mean values derived from both kidneys. All 24 features belonged to high-order features, as shown in Figure 3. Of these, 2 features were from the first-order statistics class, while the remaining were texture features. Further details regarding these radiomic features are provided in the Appendix A.

### 3.3. Model Evaluation and Comparison

The GP classifier demonstrated the best classification performance. The ROC curves of the radiologist and the GP model for the training and testing sets are presented in Figure 4a,b.

The area under the curve (AUC) of the radiologist was significantly lower than that of the GP radiomics model in the training set [0.849 (95% confidence interval (CI), 0.831–0.866) versus 0.570 (95% CI, 0.546–0.593), *p* < 0.05] and in the testing set [0.79 (95% CI, 0.748–0.831) versus 0.575 (95% CI, 0.527–0.622), *p* < 0.05] (Table 2). The sensitivity of the radiologist was lower than that of the radiomics analysis (*p* < 0.05), while the specificity was higher (*p* < 0.05) (Table 2).

In Figure 5, the calibration curves for both the training (c) and testing sets (d) show strong concordance between the predicted and actual prevalence of CKD stages 1–3. Figure 6 further evaluates clinical utility using decision curve analysis for the training (e) and testing sets (f).

## 4. Discussion

The identification of CKD is a prolonged process for patients. CT technology is widely used in the evaluation of renal diseases, including CKD. However, its applications are predominantly limited to manual analysis, which relies heavily on the personal experience of professional physicians and lacks standardization. The results of the current study demonstrate that the AUC and sensitivity of radiologists in identifying early CKD were only 0.570–0.575 and 0.196–0.200, respectively, in the training and testing sets. Nearly half of the patients could not be accurately identified. This finding aligns with previous ultrasound studies, which revealed that professional radiologists had low sensitivity for the identification and grading of CKD, with the issue becoming more pronounced as the disease grade decreased [16].

The current results suggest that the constructed radiomics features model can effectively differentiate early CKD from healthy kidneys, achieving an AUC of 0.79–0.849 and a sensitivity of 0.709–0.750. The AUC and sensitivity of the GP model were 21.5% to 27.9% and 50.9% to 55.4% higher, respectively, than those of the radiologist in both the testing and training sets. A previous ultrasound study also demonstrated that the diagnostic accuracy and sensitivity of the radiomics model for CKD stages 1–3 were significantly higher than those of senior radiologists [16].

For model validation, the calibration curves show strong alignment with the dotted line, which represents an ideal model. This indicates that the GP model exhibits no significant deviation from a perfect fit, suggesting that its predictions closely mirror the true prevalence of CKD in the given datasets. Furthermore, the GP model outperforms traditional diagnostic methods used by physicians. The clinical decision curve plots net benefit on the *y*-axis, calculated by balancing the gains from true positives against the costs of false positives. The GP model consistently demonstrates a higher net across the entire range of threshold probabilities. This superior performance in both clinical decision-making and practical application underscores the model’s value in enhancing diagnostic accuracy for CKD stages 1–3 compared to conventional radiologist diagnoses.

From the perspective of feature extraction, healthy kidney tissues exhibit higher values in features such as Inverse Variance (speckle noise), Large Dependence Emphasis (wavelet-LLH), 90th Percentile (wavelet-LLH), and Dependence Entropy (wavelet-LLH) on CT images. This indicates that healthy kidneys have more homogeneous texture, uniform regions, and higher intensity values on CT images. Inverse Variance reflects the uniformity in pixel intensity, with higher values suggesting more consistent tissue. Large Dependence Emphasis indicates the presence of extensive, similar-intensity regions, which are characteristic of healthy, undamaged kidney tissue. The 90th Percentile value denotes the range of higher-intensity pixels, implying robust and dense renal parenchyma. Additionally, higher Dependence Entropy implies a complex and varied texture pattern, which is characteristic of normal kidney function and structure. These features contrast sharply with the lower values typically observed in chronic kidney disease, which is marked by fibrosis, atrophy, and overall tissue degradation. The reduced values in these features for CKD kidneys reflect less homogeneous texture, smaller homogeneous regions, and lower intensity values due to pathological changes such as fibrosis and atrophy. Lower Large Dependence Emphasis can be attributed to the diminished prevalence of large, homogeneous regions within diseased kidneys, a hallmark of chronic pathological changes such as scarring and loss of functional renal parenchyma.

In contrast, diseased kidney tissue exhibits higher values in features such as Contrast (wavelet-LLH), Gray Level Non-Uniformity (discrete gaussian), Correlation (wavelet-HHL), and Imc2 (wavelet-HLH) on CT images. This signifies that CKD kidneys exhibit more pronounced heterogeneity and irregularity in texture. Higher Contrast values reflect significant variations in pixel intensity, highlighting the uneven distribution of fibrotic and atrophic areas within the kidney tissue. Gray Level Non-Uniformity measures the variability of gray levels in the image, with higher values suggesting a lack of uniformity and increased structural degradation. Furthermore, the Correlation (wavelet-HHL) feature, which captures the linear dependency of gray levels in specific orientations, shows higher values in CKD kidneys, implying disrupted and irregular structural patterns. The Imc2 (wavelet-HLH) feature, which quantifies the complexity and entropy of the image, also exhibits elevated values in diseased kidneys, indicating a more chaotic and disordered tissue architecture. These elevated values in CKD contrast sharply with the uniform and organized texture characteristics of healthy kidneys, underscoring the extent of damage and pathological changes present in CKD.

These characteristic alterations in CKD can be interpreted based on pathological studies. A common pathological change in CKD is renal fibrosis, defined as excessive accumulation of extracellular matrix (ECM) produced by cells deposited after activation and expansion. This process affects all parts of the kidney and is referred to as glomerulosclerosis, tubulointerstitial fibrosis, and arteriosclerosis of the arteries and arterioles [17]. This massive pathological proliferation of the ECM involves numerous cytokines and leads to scarring and hardening of the kidney tissue, resulting in a heterogenic appearance of the kidney [18]. This heterogeneity and hardness alter texture and other features (such as increased gray contrast, complexity, and non-uniformity) compared to healthy kidneys. The radiologists’ AUC was marginally higher in the test set (0.575) than in the training set (0.570). This minor difference (Δ = 0.005) is likely due to random variation rather than overfitting, as radiologists rely on clinical expertise rather than data-driven learning. The train–test split was applied for fair comparison with models.

This study represents the first attempt to investigate early CKD using NCCT entire renal parenchyma analysis. A previous prospective cohort study on classifying and predicting radiation-induced CKD also identified 90th Percentile and Dependence Entropy as critical features based on entire kidney enhanced CT images [10]. A small-sample ultrasound study performed texture analysis on the entire kidney, cortex, and medulla of CKD patients and HC, finding that the most accurate results were obtained from the entire kidney and the cortex region [19]. Our results suggest that higher-order texture features can identify diseased kidneys earlier than radiologists, supporting the optimization of the diagnostic process. The radiomics model can significantly enhance the detection rate of CKD stages 1–3 without biochemical examination and guide further clinical evaluation and treatment.

We recognize that this study has several limitations. The retrospective nature of the study and the use of data from a single center may have introduced certain biases. In the future, external validation will be conducted alongside large-sample, multi-center studies to further evaluate predictive performance of the model. Healthy subjects in this study were those with no history or any symptoms of kidney disease; however, only some underwent eGFR testing. Ideally, eGFR measurements should have been performed for all participants in this group.

This study also offers insights and directions for further enhancing the use of CT imaging in the diagnosis of CKD. Future research directions include investigating whether the radiomics model can serve as an effective predictor for the etiology and pathological changes of different CKD stages, and whether renal corticomedullary segmentation based on deep learning can achieve better performance than whole-kidney radiomics in predicting and classifying early CKD. Further follow-up studies could obtain more comprehensive information from CT images and integrate it with other clinical indicators closely associated with CKD to construct a more robust and comprehensive model.

## 5. Conclusions

This study investigates the diagnostic potential of radiomic features extracted from non-contrast CT (NCCT) images, compared with radiologists’ assessments, in differentiating between early-stage chronic kidney disease (CKD stages 1–3) and normal kidney. Conventional empirical qualitative evaluations by professional radiologists showed only near-random diagnostic performance. In contrast, machine learning methods transformed imperceptible biological signals into quantifiable biomarkers via radiomics. Our approach extracted 2264 high-dimensional features (shape, texture, wavelet) from automatically segmented kidney VOIs using VB-net. Key algorithms identified features reflecting early pathophysiology. And the Gaussian process classifier integrated these weakly correlated features into a robust predictive model, outperforming radiologists by shifting from subjective morphology-based assessment to objective, quantitative analysis of latent biomolecular signatures. This paradigm highlights radiomics’ ability to uncover “invisible” disease markers beyond human vision, offering transformative potential for early CKD diagnosis.

The NCCT-based radiomics model demonstrates significant clinical utility by enabling non-invasive, early diagnosis of CKD stages 1–3, outperforming radiologist assessments. This approach minimizes human error and improves diagnostic consistency through automated segmentation and advanced machine learning techniques.

## Figures and Tables

**Figure 1 bioengineering-12-00454-f001:**
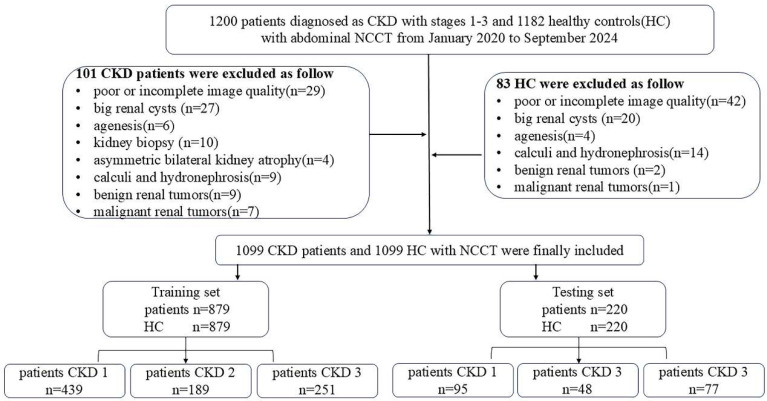
Inclusion flowchart and dataset distribution.

**Figure 2 bioengineering-12-00454-f002:**
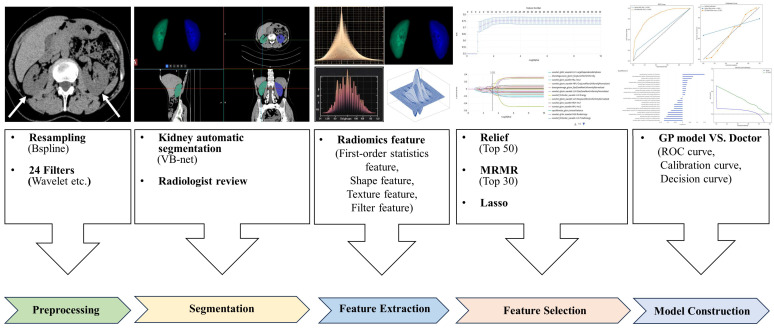
Framework for radiomics analysis based on non-contrast CT images for predicting chronic kidney disease.

**Figure 3 bioengineering-12-00454-f003:**
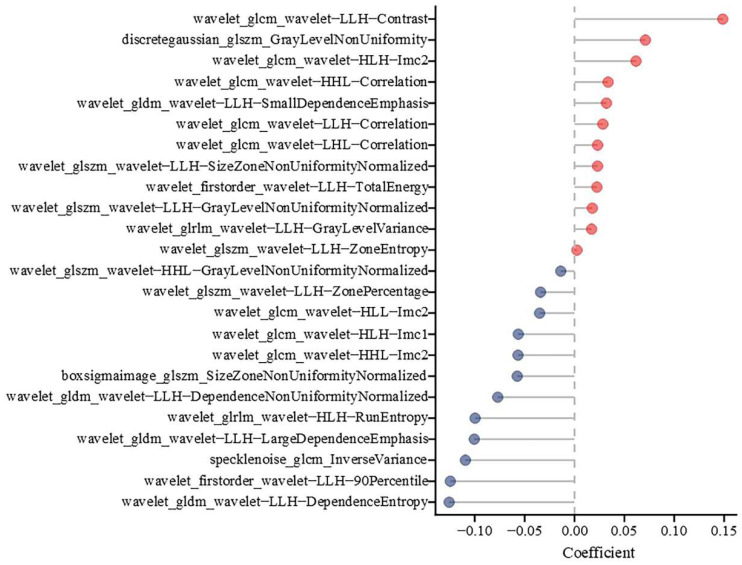
Selected features from the kidney and their LASSO regression coefficients. The red endpoints indicate features positively correlated with CKD, while the blue endpoints represent features negatively correlated with CKD.

**Figure 4 bioengineering-12-00454-f004:**
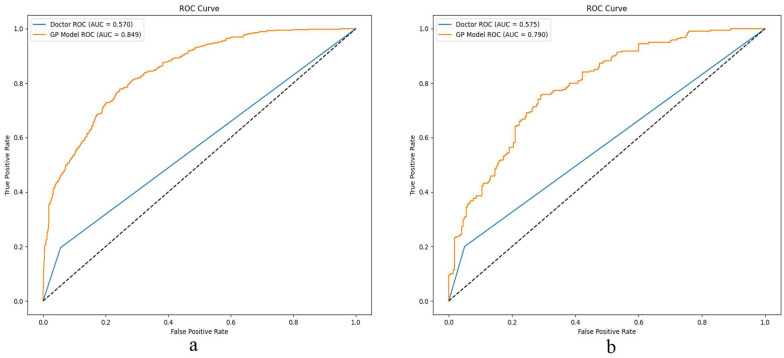
ROC curves of Gaussian process (GP) model and radiologist in the training (**a**) and testing sets (**b**).

**Figure 5 bioengineering-12-00454-f005:**
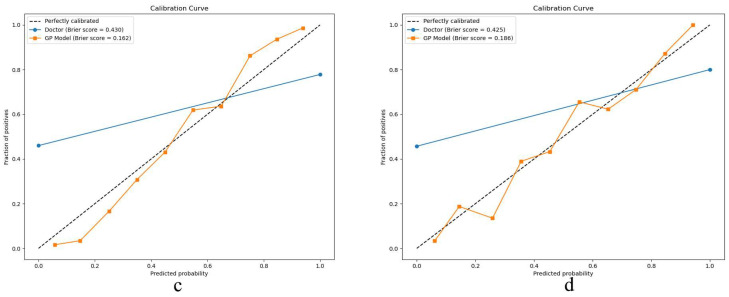
Calibration curve in the training (**c**) and testing sets (**d**).

**Figure 6 bioengineering-12-00454-f006:**
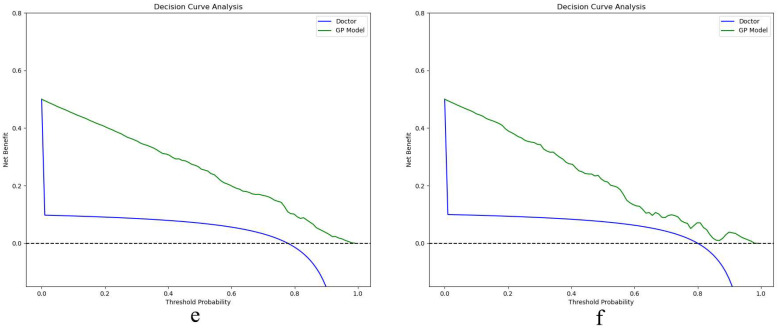
Decision curve in the training (**e**) and testing sets (**f**).

**Table 1 bioengineering-12-00454-t001:** Basic Information of the training and testing sets.

Characteristics		Training Set			Testing Set			*p*-Inter
		HC	Patients	*p*-Intra	HC	Patients	*p*-Intra	
Patient Age (Median [Q1, Q3])		50.0 (38.0, 60.0)	50.0 (37.0, 59.0)	0.334	49.0 (37.8, 60.3)	52.5 (42.0, 62.0)	0.158	0.259
Patient Sex (%)	Male	438 (49.8)	476 (54.2)	0.070	113 (51.4)	112 (50.9)	0.924	0.748
	Female	441 (50.2)	403 (45.8)		107 (48.6)	108 (49.1)		

HC, Healthy controls.

**Table 2 bioengineering-12-00454-t002:** Performance of Gaussian process model and radiologist in the training and testing sets.

Comparison	Cutoff	AUC (95%CI)		Sensitivity		Specificity		Accuracy		Precision		f1 Score	
		Training	Testing	Training	Testing	Training	Testing	Training	Testing	Training	Testing	Training	Testing
GP model	0.5000	0.849 (0.831–0.866)	0.790 (0.748–0.831)	0.750	0.709	0.770	0.732	0.760	0.720	0.765	0.726	0.757	0.717
Radiologist	0.5000	0.570 (0.546–0.593)	0.575 (0.527–0.622)	0.196	0.200	0.944	0.950						
Z Statistics		25.063	8.157										
*p* value		<0.0001 *	<0.0001 *										

95% CI, 95% confidence interval; AUC, area under the curve. * *p* value was calculated for the difference between Gaussian process model and the radiologist using the DeLong test in training and testing set.

## Data Availability

Due to the data being part of ongoing research or technical/time limitations, the dataset in this paper is temporarily unavailable. Requests for the dataset should be directed to the corresponding author.

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
