# Peer review of "Radiomics Analysis of Whole-Kidney Non-Contrast CT for Early Identification of Chronic Kidney Disease Stages 1–3"

_bioengineering, 2025, doi:10.3390/bioengineering12050454_

Round 1
Reviewer 1 Report
Comments and Suggestions for Authors
This study aims to evaluate the potential of radiomics-based quantitative features extracted from traditional non-contrast computed tomography (NCCT), combined with machine learning techniques, in differentiating CKD stages 1-3 from healthy controls.
The paper is very interesting, I have a few observations:
1. Check for typos: for example, The title in the pdf document reports an extra R (R Radiomics), line 197-198 "verus" instead of "versus"...
2. Cohen's kappa coefficient should be asssessed to measure inter-rater reliability of the two attending physicians.
3. For a better readibility, means and percentages should report only one decimal (ie. table 1 50.0 instead of 50.000 for age).
4. for the radiologist group ,you have a slight higher ROC in the test set as compared to the training set, causes should be discussed.
Author Response
Dear Reviewers,
First and foremost, I would like to express my sincere gratitude for the thorough review and insightful comments provided on our manuscript. Those comments are all valuable and very helpful for revising and improving our paper, as well as the important guiding significance to our researches. We have carefully considered each point raised and have made the following revisions to our manuscript.
Reviewer 1
Comments 1: Check for typos: for example, The title in the pdf document reports an extra R (R Radiomics), line 197-198 "verus" instead of "versus"...
Response 1: Thank you for pointing this out. We agree with this comment. Therefore, the modifications have been incorporated in both the title and the main text (lines 198-199), with all changes clearly highlighted.
Comments 2: Cohen's kappa coefficient should be assessed to measure inter-rater reliability of the two attending physicians.
Response 2: Thank you for pointing this out, we agree with this comment. The results of Cohen's kappa agreement test between the two attending radiologists have been added to the supplementary table1. The overall kappa value was 0.761, with values of 0.761 and 0.762 for the training and test sets respectively, indicating substantial inter-rater agreement between the two radiologists.
Comments 3: For a better readibility, means and percentages should report only one decimal (ie. table 1 50.0 instead of 50.000 for age).
Response 3: Thank you for pointing this out, we agree with this. The corresponding modifications have been incorporated into Table 1 in the text.
Comments 4: for the radiologist group, you have a slight higher ROC in the test set as compared to the training set, causes should be discussed.
Response 4: Thank you for your thorough review and valuable comments. Regarding the slightly higher ROC of the radiologist group in the test set compared to the training set (Training AUC 0.570 vs. Testing AUC 0.575), we provide the following clarifications: Radiologists' diagnostic performance fundamentally differs from AI models. Their expertise derives from clinical experience rather than data fitting. The train-test split was implemented solely for equitable comparison. The minimal AUC difference (Δ=0.005) likely reflects random variation in data distribution rather than overfitting or performance improvement. The modifications have been incorporated in the discussion section (page 10, lines 286-290).
Reviewer 2 Report
Comments and Suggestions for Authors
The work concerns the modern direction of development of medical information technologies. The application of the latest machine learning technologies is without a doubt appropriate and effective in the tasks of medical diagnosis of CKD. However, I suggest that the authors improve their article by paying attention to the following:
1) The article pays little attention to the detailed analysis of machine learning models and methods. Reasonable choice of these technologies.
2) It is not clear which radiomic features are the most informative?
3) Why do professional radiologists almost "not see" defects on images? What do machine methods "see" in this case?
4) It is necessary to significantly expand and supplement the conclusions of the article.
5) Why did the authors not compare the accuracy and reliability characteristics of the machine learning methods proposed in the work with similar methods used by other authors?
Author Response
Dear Reviewers,
First and foremost, I would like to express my sincere gratitude for the thorough review and insightful comments provided on our manuscript. Those comments are all valuable and very helpful for revising and improving our paper, as well as the important guiding significance to our researches. We have carefully considered each point raised and have made the following revisions to our manuscript.
Comments 1: The article pays little attention to the detailed analysis of machine learning models and methods. Reasonable choice of these technologies.
Response 1: Thank you for pointing this out. Our study encompassed a comprehensive evaluation of multiple machine learning models and methodologies. While only the optimal model's results were initially presented in the manuscript, we have now incorporated the complete comparative analysis of all models in the Supplementary Table 2 in response to your valuable suggestion.
Comments 2: It is not clear which radiomic features are the most informative?
Response 2: Thank you for pointing this out. Figure 3 and the accompanying supplementary table3 present the selected optimal features from the kidney and their LASSO regression coefficients. The red endpoints indicate features positively correlated with CKD, while the blue endpoints represent features negatively correlated with CKD.
Comments 3: Why do professional radiologists almost "not see" defects on images? What do machine methods "see" in this case?
Response 3: Thank you for pointing this out. Professional radiologists may "not see" early-stage CKD defects on non-contrast CT (NCCT) images due to early CKD stages (1-3, eGFR ≥60 mL/min/1.73m²) lack overt anatomical alterations (e.g., volume atrophy or morphological abnormalities) visible to the human eye. These subclinical functional changes (e.g., declining glomerular filtration rates) are undetectable via traditional qualitative evaluation, as evidenced by the near-random diagnostic performance of radiologists (AUC=0.570–0.575). In contrast, machine learning methods "see" by translating imperceptible biological signals into quantifiable biomarkers via radiomics. Our approach extracted 2,264 high-dimensional features (shape, texture, wavelet) from automatically segmented kidney VOIs using VB-net. Key algorithms (Relief, MRMR, LASSO) identified features reflecting early pathophysiology. And the Gaussian process classifier integrated these weakly correlated features into a robust predictive model (AUC=0.790–0.849), outperforming radiologists by shifting from subjective morphology-based assessment to objective, quantitative analysis of latent biomolecular signatures. This paradigm highlights radiomics’ ability to uncover "invisible" disease markers beyond human vision, offering transformative potential for early CKD diagnosis.
Comments 4: It is necessary to significantly expand and supplement the conclusions of the article.
Response 4: We sincerely appreciate your valuable suggestion regarding the expansion of our conclusions. In response to your comment, these modifications are highlighted in the revised manuscript conclusions section (page 10, line 318-331).
Comments 5: Why did the authors not compare the accuracy and reliability characteristics of the machine learning methods proposed in the work with similar methods used by other authors?
Response 5: Thank you for pointing this out. while numerous radiomics studies have investigated chronic kidney disease using ultrasound imaging, CT-based approaches remain scarce with significant variations in research objectives, methodologies, and data modalities. The present study addresses a critical clinical challenge, and to our knowledge, represents the first investigation employing this specific approach. In the Discussion section, we provide comparative analysis with related studies, while following Table systematically summarizes the methodological characteristics of comparable publications."
Team |
mode |
Method/model |
Subjects |
AUC |
Our study |
Non-contrast CT |
Radiomics/GPmodel |
Healthy controls, CKD (stage1-3) |
radiomics model vs. physician((Training/ testing ) 0.849/0.790 vs. 0.570/0.575 |
Yoon Ho Choi[1] |
Non-contrast CT |
shape Features |
CKD (stage3-5) |
Different shape features ranged between 0.51- 0.86 |
Shuyuan Tian[2] |
ultrasound |
Deep learning radiomics |
Healthy controls, CKD (stage1-5) |
radiomics model vs. physician Non-CKD vs. CKD 0.918vs 0.869 CKD stage1 0.781 vs. 0.506 CKD stage2 0.880 vs. 0.586 CKD stage3 0.905 vs. 0.796 |
Muditha S Bandara[3] |
ultrasound |
Radiomics/ SVM |
Healthy controls, CKD (stage2-4) |
radiomics model 0.88 |
- Choi YH, Jo S, Lee RW, Kim JE, Paek JH, Kim B, Shin SY, Hwang SD, Lee SW, Song JH, Kim K. Changes in CT-Based Morphological Features of the Kidney with Declining Glomerular Filtration Rate in Chronic Kidney Disease. Diagnostics (Basel). 2023 Jan 22;13(3):402. doi: 10.3390/diagnostics13030402. PMID: 36766507; PMCID: PMC9914455.
- Tian S, Yu Y, Shi K, et al. Deep learning radiomics based on ultrasound images for the assisted diagnosis of chronic kidney disease[J]. Nephrology (Carlton), 2024, 29 (11): 748-757.
- Bandara MS, Gurunayaka B, Lakraj G, Pallewatte A, Siribaddana S, Wansapura J. Ultrasound Based Radiomics Features of Chronic Kidney Disease. Acad Radiol. 2022 Feb;29(2):229-235. doi: 10.1016/j.acra.2021.01.006. Epub 2021 Feb 12. PMID: 33589307.
Reviewer 3 Report
Comments and Suggestions for Authors
The manuscript by Zhang et al. is clear and well-structured.
It looks at an important medical problem: finding early-stage CKD using non-contrast CT and radiomics. The study includes a large number of patients and follows a solid machine learning approach. The model performs much better than radiologists, both in training and test sets.
The methods are explained clearly, especially how the kidneys were segmented.
One thing needs to be explained better: the paper does not say how the cut-off values were chosen for sensitivity, specificity, and AUC. This is important for understanding how well the model works.
The discussion mentions the study’s limits, like the need for outside validation.
There seems to be a mistake in the title: “R Radiomics Analysis” is unclear and should be corrected.
Overall, this is a strong and useful study.
Recommendation: Accept after minor revision.
Author Response
Dear Reviewers,
First and foremost, I would like to express my sincere gratitude for the thorough review and insightful comments provided on our manuscript. Those comments are all valuable and very helpful for revising and improving our paper, as well as the important guiding significance to our researches. We have carefully considered each point raised and have made the following revisions to our manuscript.
Comments 1: One thing needs to be explained better: the paper does not say how the cut-off values were chosen for sensitivity, specificity, and AUC. This is important for understanding how well the model works.
Response 1: We sincerely appreciate your valuable suggestion regarding the cutoff values. In accordance with your recommendation, we have revised Table 2 to implement the optimal cutoff values of 0.5 for sensitivity, specificity, and AUC in our analysis.
Comments 2: There seems to be a mistake in the title: “R Radiomics Analysis” is unclear and should be corrected.
Response 2: We sincerely appreciate your meticulous review and apologize for this oversight in our initial submission. The necessary corrections have been carefully implemented throughout the manuscript.